# Prevalence and Antimicrobial Susceptibility Patterns of Bacterial Pathogens in Urinary Tract Infections in University Hospital of Campania “Luigi Vanvitelli” between 2017 and 2018

**DOI:** 10.3390/antibiotics9050215

**Published:** 2020-04-28

**Authors:** Veronica Folliero, Pina Caputo, Maria Teresa Della Rocca, Annalisa Chianese, Marilena Galdiero, Maria R. Iovene, Cameron Hay, Gianluigi Franci, Massimiliano Galdiero

**Affiliations:** 1Department of Experimental Medicine, University of Study of Campania“Luigi Vanvitelli”, 80138 Naples, Italy; veronica.folliero@unicampania.it (V.F.); annalisa.chianese@unicampania.it (A.C.); marilena.galdiero@unicampania.it (M.G.); mariarosaria.iovene@unicampania.it (M.R.I.); cameronhay022@gmail.com (C.H.); 2Section of Microbiology and Virology, University Hospital Luigi Vanvitelli of Naples, 80138 Naples, Italy; pinacaputo@hotmail.it (P.C.); mariateresa.dellarocca@unicampania.it (M.T.D.R.); 3Department of Medicine, Surgery and Dentistry “Scuola Medica Salernitana”, University of Salerno, SA 84081 Baronissi, Italy

**Keywords:** urinary tract infection, uropathogen, antibiotic, antimicrobial resistance

## Abstract

Urinary tract infections (UTIs) are the most common and expensive health problem globally. The treatment of UTIs is difficult owing to the onset of antibiotic-resistant bacterial strains. The aim of this study was to define the incidence of infections, identify the bacteria responsible, and identify the antimicrobial resistance profile. Patients of all ages and both sexes were included in the study, all admitted to University Hospital of Campania “Luigi Vanvitelli”, between January 2017 and December 2018. Bacterial identification and antibiotic susceptibility testing were performed using matrix assisted laser desorption ionization-time of flight mass spectrometry (MALDI-TOF MS) and Phoenix BD. Among the 1745 studied patients, 541 (31%) and 1204 (69%) were positive and negative for bacterial growth, respectively. Of 541 positive patients, 325 (60%) were females, while 216 (39.9%) were males. The largest number of positive subjects was recorded in the elderly (>61 years). Among the pathogenic strains, 425 (78.5%) were Gram-negative, 107 (19.7%) were Gram-positive, and 9 (1.7%) were Candida species. The most isolated Gram-negative strain is *Escherichia coli* (*E. coli*) (53.5%). The most frequent Gram-positive strain was *Enterococcus faecalis* (*E. faecalis*) (12.9%). Gram-negative bacteria were highly resistant to ampicillin, whereas Gram-positive bacteria were highly resistant to erythromycin.

## 1. Introduction

Urinary tract infections (UTIs) are common human microbial diseases that affect the urinary tract—the kidneys, bladder, urethra, and prostate [1]. UTIs are widespread globally with direct and indirect social and economic effects. Moreover, these diseases are becoming an emergent cause of morbidity. It is estimated that UTIs affect about 150 million people each year in the world. The healthcare costs are over $6 billion [2]. These human diseases are second only to respiratory tract infections [3]. In the United States, UTIs lead to over 1 million medical examinations in the emergency department and cause over 100,000 hospital admissions each year. However, UTIs can be clustered into community or nosocomial acquired [4]. The first one, community-acquired urinary tract infections (CA-UTI), occur in community or following less than 48 h of hospitalization. Nosocomial urinary tract infections (N-UTI), instead, appear 48 h after hospital admission or three days after discharge [5]. The UTIs distribution in the population changes depending on age, sex, catheterization, hospitalization, and prolonged use of antimicrobials [6]. Bacteria represent the main cause of UTIs, although viruses, fungi, and parasites may be involved in the development of this infection [7,8]. Gram-negative bacteria are responsible for 90% of UTI cases, while gram-positive bacteria are responsible for the other 10%. Previous studies suggest that the most common cause of UTIs is *Escherichia coli* (*E. coli*), which represent 65–90% of infections [9,10]. Other uropathogens that cause UTIs include Enterococcus species, *Klebsiella pneumoniae* (*K. pneumoniae*), Citrobacter species, *Pseudomonas aeruginosa* (*P. aeruginosa*), and Staphylococcus coagulase negative (CoNS) [11]. The UTIs’ diagnosis is based on the analysis of the patient’s clinical symptoms and laboratory tests [11]. The reported clinical symptoms of UTI are based on the following: (i) anatomical district; (ii) the microorganisms; (iii) severity of the infection; and (iv) the patient’s immune system [12]. The following symptoms are related: (i) urinary frequency and pain upon urination, (ii) back pain, (iii) dysuria, (iv) pyuria, and (v) abdominal pain [13,14]. Sometimes, the presence of bacteria in the urinary tract can be also associated with no symptoms [15]. Cultural examination and antibiotic susceptibility testing are routine laboratory procedures. Rapid and accurate diagnosis can be useful for quick recovery and prevention of some complications, such as pyelonephritis, kidney failure, or sepsis [16]. The global problem related to uropathogens is closely linked to the emergence of antimicrobial resistance (AMR) [17]. In Europe, 9% of all prescribed antibiotics are for the treatment of UTIs [18]. Patients with UTIs have a negative outcome following the failure of antibiotic treatment, with the development of severe clinical complications [19,20]. Drug-resistant bacterial strains and the high incidence of UTIs should highlight the need for greater understanding of microorganisms that cause UTIs and their antibiotic susceptibility pattern. The aim of this study was to assess the bacterial pathogens implicated in UTIs and their antimicrobial susceptibility profile in patients admitted at the University Hospital of Campania Luigi Vanvitelli. Knowledge of the main uropathogens and related models of antibiotics susceptibility is essential to allow the optimal choice of antibiotic therapy.

## 2. Results

### 2.1. Incidence of UTIs in Studied Patients

In the present study, 1745 urinary specimens were examined. UTIs are diagnosed based on patient’s clinical symptoms, presence of leukocytes, and bacteria in the urine. Of 1745 samples, 541 (31%) were positive for growth of pathogenic strains, while 1204 (69%) were negative (Table 1). Among the 541 pathogenic isolates, 107 (19.7%) were Gram-positive, 425 (78.5%) were Gram-negative, and 9 (1.7%) were Candida species (Table 1). With reference to gender, the positive cultures for women and men were 325 (60.1%) and 216 (39.9%), respectively (Table 1). Regarding age distribution of infection, the major part of positives was found in the elderly (>61 years) (45.5%), followed by late adulthood (46–60 years) (19%), young adults (19–45 years) (13.1%), infants (<1 years) (9.9%), adolescents (13–18 years) (5.9%), late childhood (late (6–12 years) (3.5%), and early childhood (2–5 years) (3.1%). The female to male ratio was higher in the age group 13–18 years (F/M = 2.6), while it was lower in the age group 2–5 years (F/M = 0.42) (Table 1). Most of the analyzed positive patients are admitted to the department of Internal Medicine, Urology, Geriatrics, and Pediatrics, according to age distribution. Bacterial species belonging to 13 genera were isolated and identified by 541 positive cultures. For Gram-positive bacteria, our data showed that *E. faecalis* was the most common isolated bacterium (12.9%), followed by *Enterococcus faecium* (*E. faecium*) (2%), CoNS (2%), *Streptococcus agalactiae* (*S. agalactiae*) (1.3%), *Staphylococcus aureus* (*S. aureus*) (1.1%), *Bacillus cereus* (*B. cereus*) (0.4%), and *Streptococcus gallolyticus* (*S. gallolyticus*) (0.2%) (Figure 1a). For Gram-negative bacteria, our data exhibited that *E. coli* was the most frequently isolated bacterium (53.5%), followed by *K. pneumoniae* (7%), *P. aeruginosa* (5.5%), *Proteus mirabilis* (*P. mirabilis*) (3.3%), Citrobacter species (3.1%), *Acinetobacter baumannii* (*A. baumannii*) (2.8%), *Enterobacter cloacae* (*E. cloacae*) (1.5%), *Klebsiella oxytoca* (*K. oxytoca*) (0.7%), *Morganella morganii* (*M. morgani*) (0.6%), *Klebsiella aerogenes* (*K. aerogenes*) (0.2%), and Salmonella species (0.2%) (Figure 1b).

### 2.2. Prevalence of Antimicrobial Resistance among Identified Uropathogens

In the present study, the antimicrobial resistance patterns of Enterococcus spp., *S. aureus*, CoNS, *E. coli*, *A. baumannii*, Citrobacter spp., Klebsiella spp., *P. mirabilis*, and *P. aeruginosa* were identified. The antimicrobial resistance profile is shown in Figure 2 and Figure 3. All isolated bacteria exhibited a high rate of resistance to the analyzed antibiotics. The data show that, among the Gram-positive bacteria causing UTIs, the most frequent and resistant species is represented by Enterococcus spp. This strain had a resistance greater than 97.5% to five antibiotics: erythromycin, fusidic acid, cefoxitin, gentamicin, trimethoprim-sulfamethoxazole, and clindamycin. They showed a susceptibility of lower than 2.5% to linezolid, teicoplanin, and vancomycin. Enterococcus spp., *S. aureus*, and CoNS exhibited an important resistance to erythromycin of 98.8%, 81.8%, and 83.3%, respectively. *S. aureus* was 100% susceptible to fusidic acid, gentamicin, linezolid, teicoplanin, vancomycin, and daptomycin. CoNS isolates were not susceptible as *S. aureus*. In fact, they were 100% sensitive only to trimethoprim-sulfamethoxazole, vancomycin, and daptomycin (Figure 2). Our results established that, among the Gram-negative strains, *E. coli* was the most frequent strain, but *A. baumannii* was the most resistant bacterium. The latter showed 100% resistance to ampicillin, amoxicillin/clavulanate, cefotaxime, cefuroxime, and fosfomycin, followed by amikacin, levofloxacin, gentamicin, imipenem (all 93.3%), and tobramycin (80%). Most *E. coli* isolates were resistant to ampicillin (66.2%) and less resistant to Imipenem (0.7%). Citrobacter and Klebsiella spp. had 100% resistance to ampicillin. *P. mirabilis* had significant rates of resistance to gentamicin (83.3%). *P. aeruginosa* exhibited a resistance greater than 96.8% to four antibiotics: ampicillin, amoxicillin/clavulanate, cefotaxime, and cefuroxime (Figure 3).

## 3. Discussion

The present study determines the incidence of UTIs, evaluates the pathogens involved in the infection, and estimates their sensitivity profile. Out of 1745 urinary specimens collected during this study, 541 (31%) patients had urine samples with a significant bacteriuria. Our data were similar to results of a study conducted in Saudi Arabia (32.6%) in terms of frequency [21]. Lower frequency of infections was observed at National Hospital Abuja in Nigeria (13.1%) [22]. The majority of patients with UTI were female (60.1%), consistent with prior studies [23,24]. Female patients, in fact, were more predisposed to the urinary infection because of their genital anatomy. Regarding age, the elderly group (45.5%) had a higher incidence of infection owing to some of the following factors: (i) urinary tract abnormalities, (ii) urinary and fecal incontinence, (iii) immune response decrease, (iv) disability, (v) diabetes, and (vi) prostate alteration in men and hormonal changes in female [25]. Our data showed that, of the 541 pathogenic isolates, 19.7% were Gram-positive, 78.5% were Gram-negative, and 1.7% were Candida species. The bacterial identification showed that, among Gram-positive bacteria, *E. faecalis* (12.9%) was the most isolated strain, while *S. gallolyticus* (0.2%) was the least frequent. Among the Gram-negative bacteria, *E. coli* (53.5%) was the most detected one, while Salmonella spp. (0.2%) was the least isolated [26]. The frequency of bacteria isolates was similar to other studies in different countries. In South America, *E. coli* was the most frequently isolated organism and was responsible for 39.7% of UTI cases, followed by Enterococcus spp. (11.5%) [27]. Likewise, in China, *E. coli* was the most isolated uropathogen, involved in 66.01% of UTIs cases, followed by Enterococcus spp. (5.91%) [28]. Bacterial resistance patterns showed Enterococcus spp. as the most resistant strain to different tested antibiotics: erythromycin, fusidic acid, cefoxitin, gentamicin, trimethoprim-sulfamethoxazole, and clindamycin. On the other hand, *S. aureus* was the most Gram-positive sensitive strain to fusidic acid, gentamicin, linezolid, teicoplanin, vancomycin, and daptomycin. Our data showed that erythromycin is inefficient in the antibiotic treatment of UTIs caused by Gram-positive bacteria. Resistance rates of trimethoprim-sulfamethoxazole (78.4%) and gentamicin (84.2%) are worrying. Similar data were obtained from a study conducted in Iraq by Al-Nashbandi and colleagues [29]. The inverse results were detected in a study of Bitew et al. They reported 42% and 7% of Gram positive bacteria resistant to trimethoprim-sulfamethoxazole and gentamicin, respectively. [30]. Concerning Gram-negative bacterial strains, *A. baumannii* was identified as the most resistant bacterial isolate. This strain had exhibit 100% resistance to ampicillin, amoxicillin/clavulanate, cefotaxime, cefuroxime, and fosfomycin. Citrobacter spp. was the most sensitive, particularly to amikacin, levofloxacin, gentamicin, imipenem, and trimethoprim-sulfamethoxazole. Gram-negative strains showed high resistance to Ampicillin (72%), according to the Campania region antibiotic-resistant report. A similar result was reported in Ethiopia, where 78% of Gram-negative bacteria were resistant to ampicillin [30]. Trimethoprim-sulfamethoxazole, gentamicin, and ampicillin represent some of the first-line empiric treatments, routinely used for the treatment of UTIs [31]. In almost all cases of UTIs, empirical antibiotic treatment begins before the urine culture results. Therefore, misuse of antibiotic treatment increases antibiotic resistance among uropathogens. Several studies highlight the need to use antibiotics properly, in order to overcome the antibiotic resistance problem [32]. Our data underlined that imipenem could be used for treatment of Gram-negative strains. Studies conducted in our region reported that imipenem resistance was the lowest compared with other antibiotics. In contrast, penicillin resistance rate was the highest with an upward trend. Linezolid, teicoplanin, vancomycin, and daptomycin could be the proper therapies for the Gram-positive isolates, in line with our regional reports [33]. The study population included hospitalized patients, who may be suffering from co-infections owing to multi-drug resistant bacterial strains. Treatment with these antibiotics could help solve the co-infection problems, including UTIs, which usually afflict this type of patient. In awareness of the antimicrobial resistance problem in hospital and care settings, our study will influence the choice of empirical therapy for urinary tract infection. The reported antibiotic susceptibility profiles focused attention on dysfunctions of prescribing pathways and suggest improvements in the management and control of urinary infections. We suggest that empirical antibiotic therapy should be based on knowledge of the located epidemiological trend. Our study promotes information on the current situation in our university hospital, in order to establish new guidelines for the correct use of antibiotics.

## 4. Materials and Methods

### 4.1. Sample Collection

A total of 1745 urinary specimens were collected from patients of University Hospital of Campania “Luigi Vanvitelli” in Naples between January 2017 and December 2018. Midstream specimens of urine (MSU) were delivered to the bacteriology laboratory and processed.

### 4.2. Inclusion and Exclusion Criteria

Inclusion criteria were as follows: (i) study participants included patients aged 0 to 99; (ii) all patients showed clinical evidence with one or more symptoms of an UTI (e.g., dysuria, frequency, hesitation, urgency, pain); and (iii) the culture was positive when the bacterial count was greater than 105 CFU/mL in the urine at mid flow. Exclusion criteria were as follows: (i) patients with urinary catheter were not part of our study; and (ii) the culture was negative when the bacterial count less than 103 CFU/mL in medium flow urine.

### 4.3. Bacterial Culture and Identification

The samples were spread on blood agar and MacConkey and Sabouraud Glucose agar medium, and incubated overnight at 37 °C. When the growth of two or more bacterial species was observed, the samples were regarded as contaminated (exclusion criteria). Urinary cultures were negative if the number of colony forming units per mL (CFU/mL) was less than 103 (exclusion criteria). Bacteriuria was defined by the number of over 105 CFU/mL and monomorphic growth (inclusion criteria). In this instance, bacterial identification and antimicrobial sensitivity test were executed [34]. Bacterial identification was obtained via matrix assisted laser desorption ionization-time of flight mass spectrometry (MALDI-TOF MS) (Bruker Dal-tonics, Bremen, Germany). Two colonies from a culture agar plate was distributed on a MSP 96 MALDI-TOF (Bruker Dal-tonics, Bremen, Germany). Each well was coated with 1 μL of matrix solution (saturated solution of alpha-cyano-4-hydroxycinnamic acid in 50% of acetonitrile and 2.5% of trifluoroacetic acid) (Bruker Dal-tonics, Bremen, Germany) and dried for 5 min. The obtained spectra were imported into MALDI BioTyper 3.0 software (Bruker Dal-tonics, Bremen, Germany) and evaluated through standard pattern matching compared with the main spectra. A score higher than 2 was allowed for the identification of the species [35,36].

### 4.4. Antibiotic Susceptibility Test

The Phoenix BD (Becton Dickinson, NJ, USA) was used to confirm the identification obtained through MALDI TOF MS and to perform antibiotic susceptibility tests. In short, the identification broth (ID) was inoculated with pure bacterial colonies and adjusted to the McFarland (McF) of 0.5, using a Phoenix (Becton Dickinson, NJ, USA). A 25 μL volume of standardized ID broth suspension was added to the Phoenix broth (Becton Dickinson, NJ, USA), which was previously integrated with a drop of Phoenix indicator (Becton Dickinson, NJ, USA). Indicator and broth were loaded into the Phoenix panels, which were sealed, registered, and deposited in the Phoenix device. The results were explained using Epicenter software version 7.22A (Becton Dickinson, NJ, USA) after 16 h of incubation [37,38]. The examined antimicrobials in this study were ampicillin, amoxicillin/clavulanic acid, amikacin, cefotaxime, cefuroxime, fosfomycin, gentamicin, imipenem, levofloxacin, trimethoprim/sulfamethoxazole, tobramycin, piperacillin/tazobactam, erythromycin, fusidic acid, cefoxitin, linezolid, teicoplanin, vancomycin, clindamycin, oxacillin, and daptomycin.

### 4.5. Data Analysis

Data were analyzed using IBM SPSS software (version 22.0; IBM SPSS Inc., New York, USA) [39,40]. Descriptive statistics were computerized for the study variables like sex and pathogenic bacteria isolated from the study population. Tables show the frequency of isolated uropathogens bacteria and also compare the resistance percentage of UTI antibiotics. For the categorical variables, Chi-square test < 0.05 was considered significant.

## Figures and Tables

**Figure 1 antibiotics-09-00215-f001:**
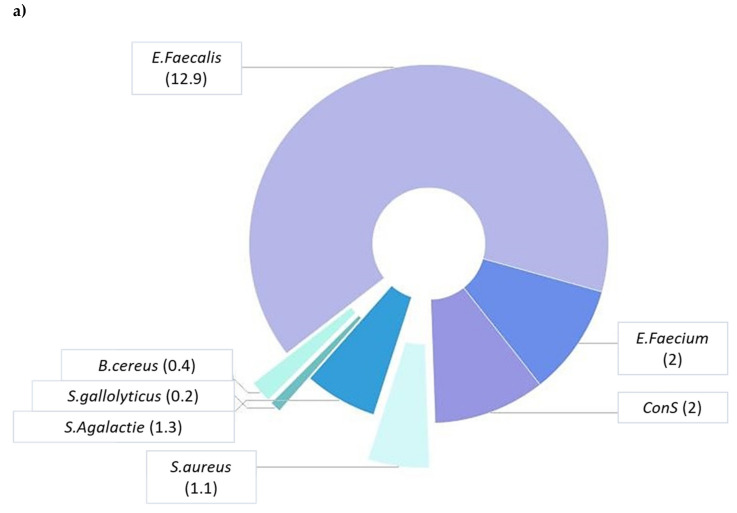
Prevalence of Gram-positive (**a**) and Gram-negative (**b**) bacteria isolated from urine samples.

**Figure 2 antibiotics-09-00215-f002:**
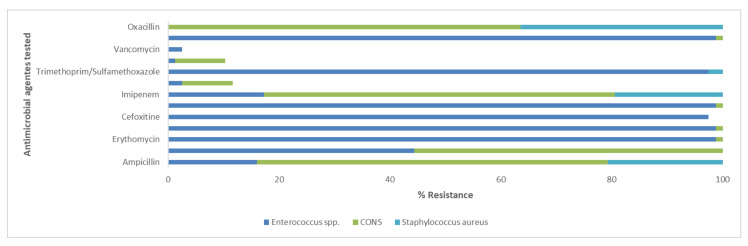
Resistant strains of most representative Gram-positive uropathogens isolated from patients at University Hospital of Campania “Luigi Vanvitelli”. Single bar refers to antimicrobial agents tested. Different colored bars indicate the resistance percentages for individual bacteria. The absence of the color in the bar means 0% of resistance for the relative strain. CoNS, Staphylococcus coagulase negative.

**Figure 3 antibiotics-09-00215-f003:**
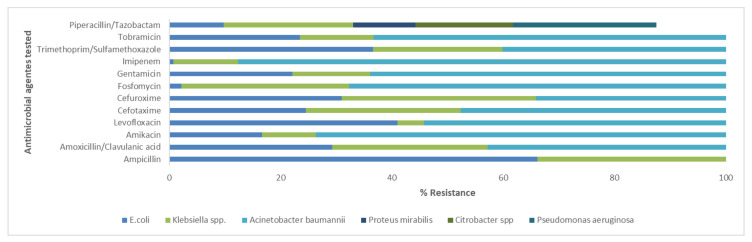
Resistant strains of most representative Gram-negative uropathogens isolated from patients at University Hospital of Campania “Luigi Vanvitelli”. Single bar refers to antimicrobial agents tested. Different colored bars indicate the resistance percentages for individual bacteria. The absence of the color in the bar means 0% of resistance for the relative strain.

**Table 1 antibiotics-09-00215-t001:** Urinary tract infections’ (UTIs) distribution of pathogenic and non-uropathogenic bacteria among tested patients in relation to gender and age.

**Character**		***n* (%)**	
No growth bacteria		1204 (69.0)	
Pathogenic bacteria		541 (31)	
Gram +		107 (197)	
Gram −		425 (78.5)	
**Gender**		***n* (%)**	
Female		325 (60.1)	
Male		216 (39.9)	
**Age Groups *n* (%)**			
	**Male**	**Female**	**Tot.**
**<1**	33 (15.3)	20 (6.2)	53 (9.9)
**2–5**	12 (5.6)	5 (1.5)	17 (3.1)
**6–12**	7 (3.2)	12 (3.7)	19 (3.5)
**13–18**	9 (4.2)	23 (7.1)	32 (5.9)
**19–45**	23 (10.6)	48 (14.8)	71 (13.1)
**46–60**	40 (18.5)	63 (19.4)	103 (19.0)
**>61**	92 (43.6)	154 (47.4)	246 (45.5)

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
