# Peer review of "Prevalence and Antimicrobial Susceptibility Patterns of Bacterial Pathogens in Urinary Tract Infections in University Hospital of Campania “Luigi Vanvitelli” between 2017 and 2018"

_antibiotics, 2020, doi:10.3390/antibiotics9050215_

Round 1

Reviewer 1 Report

The revised manuscript is considerably improved. Specific comments are as follows: 

  1. In discussion, even though Imipenem and Vancomycin (and several antibiotics) showed lower resistance in UTI, the use of these antibiotics can be difficult in clinical setting, especially in the early timing. The authors had better to address how we should do in this situation.
  2. Minor. Table 1, Figure 1 Characters are too small → please rewrite larger font. 
  3. Methods: Inclusion and exclusion criteria→ please rewrite as sentence(s).  
  4. Headings, RESULT, MATERIALS AND METHODS should be moved to appropriate places. 

Author Response

R:The revised manuscript is considerably improved. Specific comments are as follows: 

A:We are thanks to the reviewer for the critical and expert opinion that gave us the possibility to improve the manuscript. We followed all the reviewer suggestion and now the manuscript sounds better.

R.1) In discussion, even though Imipenem and Vancomycin (and several antibiotics) showed lower resistance in UTI, the use of these antibiotics can be difficult in clinical setting, especially in the early timing. The authors had better to address how we should do in this situation.

A.1) We are thanks to the reviewer for the suggestion. We added your suggestion in the discussion part (from line 175 to 178).

R.2) Minor. Table 1, Figure 1 Characters are too small → please rewrite larger font. 

A.2) We agree with the reviewer. We made the changes.

R.3) Methods: Inclusion and exclusion criteria→ please rewrite as sentence(s). 

A.3) We are thanks to the reviewer for the suggestion. We made the changes (line 191 to 196).

R.4) Headings, RESULT, MATERIALS AND METHODS should be moved to appropriate places. 

A.4) RESULT, MATERIALS AND METHODS are now in the right places.

Reviewer 2 Report

Just one minor suggestion and one query for the authors.

  • Detailed description of Figure 2 and 3 is required.
  • Is there any reason for not choosing blood and wound samples?

Author Response

R.Just one minor suggestion and one query for the authors.

A.We are thanks to the reviewer for the critical and expert opinion that gave us the possibility to improve the manuscript. We followed all the reviewer suggestion and now the manuscript sounds better.

R.1)Detailed description of Figure 2 and 3 is required.

A.1)We are thanks to the reviewer for the suggestion. We modified the text as suggested.

R.2)Is there any reason for not choosing blood and wound samples?

A.2)The surveillance related to blood and wound infections is very active in our hospital. In contrast, an effective control is required for urinary tract infections. With this study we wanted to evaluate only the prevalence of urinary tract infections, focusing on the spectrum of bacterial uropathogens and their antimicrobial susceptibility profile. Incorrect use of antibiotics in urinary tract infections cases, involves the development of multi-resistant uropathogens. Failure of antibiotic treatment can lead to the development of serious clinical complications. To avoid this, the study aims to design new guidelines to improve the treatment of urinary tract infections.

Reviewer 3 Report

As before, overall, the manuscript fits the general scope of the journal and shows a general profile of pathogens and antimicrobial susceptibility in a single hospital setting. Data that is of significant use to the local medical community. The majority of my previous comments, as well as those of the other reviewer, appear to have been addressed. 

There are a few odd editing issues that have come up during the revision process such as lines 68 with the "results" at the end of the sentence and line 177 with the "Materials and Methods" at the end of the sentence. These need to be addressed either by the authors or by the copy editor. Beyond that, the manuscript is greatly revised and seems to need just basic copy editing before publication. 

Author Response

As before, overall, the manuscript fits the general scope of the journal and shows a general profile of pathogens and antimicrobial susceptibility in a single hospital setting. Data that is of significant use to the local medical community. The majority of my previous comments, as well as those of the other reviewer, appear to have been addressed. 

We are thanks to the reviewer for the critical and expert opinion that gave us the possibility to improve the manuscript. We followed all the reviewer suggestion and now the manuscript sounds better.

R.1) There are a few odd editing issues that have come up during the revision process such as lines 68 with the "results" at the end of the sentence and line 177 with the "Materials and Methods" at the end of the sentence. These need to be addressed either by the authors or by the copy editor. Beyond that, the manuscript is greatly revised and seems to need just basic copy editing before publication. 

A.1) We are thanks to the reviewer for the suggestion. We made the changes.